# YOLOv5s-BiPCNeXt, a Lightweight Model for Detecting Disease in Eggplant Leaves

**DOI:** 10.3390/plants13162303

**Published:** 2024-08-19

**Authors:** Zhedong Xie, Chao Li, Zhuang Yang, Zhen Zhang, Jiazhuo Jiang, Hongyu Guo

**Affiliations:** 1College of Engineering and Technology, Jilin Agricultural University, Changchun 130118, China; xiezhedong@126.com (Z.X.); lc1163334300@gmail.com (C.L.); 20220263@mails.jlau.edu.cn (Z.Y.); 20220302@mails.jlau.edu.cn (Z.Z.); jiazhuojiang@gmail.com (J.J.); 2Electronic Information Technology Research Department, Jilin Province Agricultural Internet of Things Technology Collaborative Innovation Center, Changchun 130118, China

**Keywords:** YOLOv5s-BiPCNeXt, C3-BiPC module, eggplant leaves, disease detection, lightweight network

## Abstract

Ensuring the healthy growth of eggplants requires the precise detection of leaf diseases, which can significantly boost yield and economic income. Improving the efficiency of plant disease identification in natural scenes is currently a crucial issue. This study aims to provide an efficient detection method suitable for disease detection in natural scenes. A lightweight detection model, YOLOv5s-BiPCNeXt, is proposed. This model utilizes the MobileNeXt backbone to reduce network parameters and computational complexity and includes a lightweight C3-BiPC neck module. Additionally, a multi-scale cross-spatial attention mechanism (EMA) is integrated into the neck network, and the nearest neighbor interpolation algorithm is replaced with the content-aware feature recombination operator (CARAFE), enhancing the model’s ability to perceive multidimensional information and extract multiscale disease features and improving the spatial resolution of the disease feature map. These improvements enhance the detection accuracy for eggplant leaves, effectively reducing missed and incorrect detections caused by complex backgrounds and improving the detection and localization of small lesions at the early stages of brown spot and powdery mildew diseases. Experimental results show that the YOLOv5s-BiPCNeXt model achieves an average precision (AP) of 94.9% for brown spot disease, 95.0% for powdery mildew, and 99.5% for healthy leaves. Deployed on a Jetson Orin Nano edge detection device, the model attains an average recognition speed of 26 FPS (Frame Per Second), meeting real-time requirements. Compared to other algorithms, YOLOv5s-BiPCNeXt demonstrates superior overall performance, accurately detecting plant diseases under natural conditions and offering valuable technical support for the prevention and treatment of eggplant leaf diseases.

## 1. Introduction

Eggplant, also known in Latin as *Solanum melongena L.* [1], a member of the Nightshade family, is a widely grown vegetable crop that is an important part of many of the world’s best-known cuisines [2]. Enriched with abundant nutrients and displaying remarkable adaptation to a variety of climate conditions, eggplants consequently play a significant role in global vegetable production. Despite the popularity of eggplants owing to their high water content, rich vitamin and mineral composition, and dietary fiber [3], they, akin to other crops, are susceptible to a host of diseases that pose substantial impacts on both yield and quality [4]. As shown in Figure 1, a particularly prevalent disease is the brown spot disease, also termed brown rot or blight. The brown spot disease is a fungal infection instigated by the *Phomopsis vexans*, predominantly inflicting eggplant leaves, stems, and fruits, eventually forming circular brown spots with dark borders [5]. Tissues affected gradually undergo necrosis, leading to the disease spreading to neighboring leaves in severe instances. This results in the damage and subsequent shed of eggplants, thereby causing a decline in crop yield. Another common eggplant disease is Powdery mildew, specifically caused by the *Leveillula taurica* [6]. The disease is characterized by the appearance of white powdery substances on eggplant leaves, stems, and fruit surfaces. In severe instances, it leads to the molding, rotting, and detachment of leaves, with consequential impacts on photosynthesis, ultimately causing a reduction in eggplant yield. Thus, the effective management and control of these diseases are paramount to ensure the health of eggplant plants and boost yield [7]. However, traditional disease diagnostic methods often necessitate experienced farmers or plant pathologists to perform visual inspections, which amounts to time consumption and predisposes the process toward subjective errors. This drives the need for early diagnosis, timely intervention, and effective disease management, propelling the advancement of automated accurate disease detection technologies.

In recent years, computer vision technology based on deep learning has demonstrated immense potential within domains such as agriculture [8]. Deep learning techniques possess the advantage of learning complex features from input data, allowing for the accurate detection and classification of objects, thus paving the way for the possibility of automatic diagnosis and classification of eggplant diseases. Traditional plant disease detection techniques involve farmers and experts relying on their experience for visual inspections, which is an inefficient and highly subjective method [9]. With the advancements in computer and information technology, image detection technology is gradually being applied in agriculture [10].

Efficient automatic disease detection in natural scenarios remains a challenging problem in the field of plant disease diagnosis. He J. [11] proposed the MFaster R-CNN algorithm, which uses batch normalization and mixed loss functions to improve detection accuracy. Xu L. [12] proposed a wheat leaf spot disease recognition method based on deep learning. The two-stream CNN was used to extract features, the residual channel attention block was introduced to enhance the features, and the feedback block was designed to make the network learn more important features, realizing the recognition of five types of wheat leaf diseases and achieving a recognition accuracy of 99.95%. However, only five common diseases were identified, the model needed to be retrained for other diseases, and the recognition effect of multiple diseases on a leaf was not ideal. Adding attention mechanisms can enhance the model’s ability to perceive multi-dimensional information and has proven to be an effective method to overcome the interference of complex backgrounds [13]. Jiangtao Qi [14] introduced the SE attention mechanism into YOLOv5 using a self-constructed tomato leaf disease dataset, achieving an accuracy of up to 91.07% by mimicking human visual attention mechanisms to extract key features. These attention mechanisms typically consider channel or spatial information, capturing local information but failing to acquire long-range dependencies. Ye R. [15] proposed a new method called Dynamic Head, combining multiple dynamic attention mechanisms to enhance multi-scale spatial localization and multi-task perception capabilities. Researchers have made numerous attempts to mitigate the impact of background on disease detection. Zhou [16] combined FCM-KM and Faster R-CNN techniques to reduce image noise, using a dynamic population firefly algorithm and the maximum minimum distance method to optimize K-Means clustering, determining the optimal cluster size to address the local optimum problem.

Disease images captured in natural scenes contain many small lesion features that make disease detection difficult. Various studies have been carried out to detect the characteristics of minor lesions. Sheping Zhai [17] proposed an improved SSD object detection algorithm based on Dense Convolutional Networks (DenseNet) and feature fusion, called DF-SSD. It requires only half of SSD’s parameters or 1/9 of Faster R-CNN’s parameters, achieving advanced detection results for small objects and objects with specific relationships by infusing more semantic information. Ghoury [18] carried out disease detection tasks in grapes and grape leaves, distinguishing between healthy and diseased grapes and grape leaves, with the Faster-R-CNN Inception v2 model achieving a correct classification rate of 95.57% for all test images, with classification accuracy ranging from 78% to 99%. This model demonstrated high precision but the processing times were rather extensive.

Although many methods have made progress in improving detection accuracy, most of them ignore the cost of model size and detection time, which increases the difficulty of applying them to mobile terminals. Building a model that not only meets the lightweight characteristics but can also detect small lesions in natural scenes is still a problem worthy of attention. The purpose of this study is to develop an efficient natural scene disease detection model to enhance the detection ability of small lesions. The YOLOv5s-BiPCNeXt model shows high detection accuracy and speed on the eggplant leaf disease dataset. It has the characteristics of a small model and low computational cost and is suitable for deployment on edge detection devices, so as to be better applied to disease detection in natural scenes. It also aims to provide valuable technical support for the application and development of eggplant leaf disease recognition strategy in the field of precision smart agriculture.

Dataset Construction: A comprehensive dataset of eggplant leaf diseases was constructed, including images of healthy leaves, brown spot disease, and powdery mildew;Proposed Lightweight Model: A lightweight detection model, YOLOv5s-BiPCNeXt, was proposed. This model reduces network parameters and computational complexity while improving the detection accuracy of eggplant leaves. It effectively minimizes missed and false detections caused by complex backgrounds, enhancing the detection and localization of early small lesions of brown spot disease and powdery mildew;Comparative Performance: The YOLOv5s-BiPCNeXt model exhibits higher detection accuracy and speed, with lower parameter count and model size compared to five other popular deep learning models;Edge Device Deployment: The YOLOv5s-BiPCNeXt model was deployed on the Jetson Orin Nano edge detection device, with an average recognition speed of 26 FPS, meeting the requirements of real-time detection. This deployment method ensures the application of the model in actual natural scenes, can quickly and accurately detect eggplant leaf diseases, and improves the efficiency of disease management in agricultural production.

## 2. Materials and Methods

### 2.1. Data Acquisition

The dataset used in the experiment was collected at the eggplant planting base of Jilin Agricultural University, Changchun City, Jilin Province in early May 2023, as shown in Figure 2. The leaves of eggplant plants were photographed and collected using a realme GT Master Edition mobile phone, which is manufactured by Guangdong Mobile Communication Co., Ltd. in China, equipped with a rear camera featuring the SONY IMX766 sensor, produced by Sony Corporation in Japan.The camera has a maximum of 50 million pixels, so the eggplant image is captured with a resolution of 4032 × 3024 pixels.

The collection environments were indoor greenhouses and real open-air environments to ensure the diversity of data samples and improve the generalization ability of the network model. The eggplant plants used in the experiment were in the colonization stage and growing steadily.

The actual collected images of diseased plants are shown in Figure 3. The main symptoms of eggplant leaves infected by brown spot disease are as follows: small round or irregular water-soaked spots appear on the lower leaves, which gradually expand. The lesions have a dark brown edge and a gray-white center, which can be considered indicators of infection. Symptoms of eggplant leaves infected with powdery mildew include the appearance of dot-like white filaments and irregular light green–yellow spots during the early stages of infection. As the disease progresses, these develop into powdery mildew patches of varying sizes with indistinct edges. As the infection worsens, the number of powdery mildew spots increases and a noticeable powdery substance appears on the spots, which eventually merges to form large patches. In severe cases, the leaf surfaces, both upper and lower, are covered with a thin layer of powder, resembling flour. If the white powder is wiped off, the affected tissue appears decolorized, later turning yellow and drying out. With severe infection, the leaves may become wrinkled, reduced in size, or deformed and flower buds may fail to open properly. Ultimately, the entire leaf may die.

### 2.2. Image Data Augmentation and Dataset Establishment

To minimize network overfitting and enhance the model’s generalization capability [19], a series of data augmentation operations were applied to the raw data. Figure 4 shows the original images and the effects of mixed processing operations, including adding noise, rotation, mirroring, random occlusion, scaling, and brightness adjustment. Specifically, Gaussian noise was added to the original images for the following reasons: zero-mean Gaussian noise has data across all frequencies, which can effectively distort and suppress high-frequency features, thereby reducing the likelihood of model overfitting. Consequently, adding Gaussian noise to the model can improve its robustness and make it more adaptable to real-world applications. It is worth noting that noise in actual environments is often not from a single source but is composed of multiple factors. Assuming that real-world noise is the composite effect of various independent random variables with different probability distributions, the Central Limit Theorem states that as the number of noise sources increases, the normalization and convergence of these random variables tend to follow a Gaussian distribution. Therefore, the introduction of random occlusion can better simulate this complex noise, further enhancing the robustness of the model. Additionally, the use of random occlusion helps reduce the model’s tendency to overfit. Similar to adding Gaussian noise, random occlusion can effectively distort high-frequency features, thus reducing the chance of model overfitting. This is because random occlusion encourages the model to focus more on overall features rather than local details, enhancing the model’s robustness [20].

The collected eggplant leaf images were annotated according to the symptoms of infected eggplant leaves using the software LabelImg (version 1.8.5) [21]. The bounding box for the eggplant leaf disease area in the image is added and the corresponding label file is generated. Image annotation is performed as described in Figure 5.

The 980 original photos collected were divided into 784 eggplant leaf pictures in the training set, 98 eggplant leaf pictures in the verification set, and 98 eggplant leaf pictures in the test set according to the ratio of 8:1:1 using the Python 3.10 processing tool [22]. In order to better enable the model to detect the disease characteristic information of eggplant leaves and improve the detection accuracy, data augmentation was used to extend the original 784 pictures in the training set to 4533. Specifically, the number of pictures of brown spots on eggplant leaves increased to 1890, the number of pictures of powdery mildew on eggplant leaves increased to 1575, and the number of pictures of healthy eggplant leaves increased to 1468. The picture distribution information before and after enlargement is displayed in the table. After such augmentation operation, the distribution of all kinds of data becomes more uniform and is more suitable for training models. The distribution numbers of different blades are shown in Table 1.

### 2.3. YOLOv5s-BiPCNeXt

The detection model for field eggplant leaf diseases demands strict requirements for real-time performance, lightweight design, and efficiency. Considering the deployment of the improved algorithm on edge detection devices, YOLOv5s [23] was selected as the base network structure for the detection algorithm, with the model network architecture of YOLOv5s illustrated in Figure 6. Compared to newer models like YOLOv8 and YOLOv9, the primary advantage of YOLOv5s is its smaller number of parameters and model weights, along with its faster detection speed. The main components of YOLOv5s include the input layer, backbone, multi-scale feature fusion network (Neck), and the output detection head. The C3 module in the backbone network of YOLOv5s contains many residual structures, which can increase the network depth to improve detection accuracy. These modules effectively enhance the feature extraction capability by separating and crossing feature maps. The neck network consists of a feature pyramid network (FPN) and a pyramid attention network (PAN) to achieve feature enhancement and fusion of upper and lower information flows; the output detection head has three detection layers and the loss of each detection layer consists of three types of losses: positioning, classification, and confidence. The classification and confidence losses are calculated using the binary cross entropy loss function, while the positioning loss is calculated using the complete intersection of union (CIoU) loss function. Finally, the prediction box is screened by pre-setting the confidence threshold to achieve the positioning and classification of the detection target.

This study proposes the detection model YOLOv5s-BiPCNeXt for eggplant leaf diseases, which is based on the original YOLOv5s model. It aims to improve the accuracy of detecting eggplant leaf diseases in complex environments while maintaining detection speed, reducing the model computational load to facilitate deployment on edge detection devices, and significantly enhancing the model’s performance. The improvements focus on four key areas: using the lightweight MobileNeXt network [24] structure as the backbone, introducing the Efficient Multi-scale Attention (EMA) module [25], utilizing the CARAFE upsampling operator [26] to better extract disease features from eggplant leaves, and incorporating the new C3-BiPC module to increase the detection rate of small targets. Figure 7 shows the overall framework of the YOLOv5s-BiPCNeXt model presented in this study.

Firstly, replacing the backbone network of the original YOLOv5s with lightweight MobileNeXt significantly reduces the number of parameters and computational complexity, while improving the inference speed of the model, laying a foundation for the lightweight design of the model. Secondly, the neck network introduces an efficient multi-scale attention mechanism module (EMA), optimizing the network’s focus when dealing with spatial dependencies, thereby enhancing its capability to extract multi-scale features and addressing the problem of low detection efficiency for targets of different sizes. Furthermore, by using the lightweight universal upsampling operator CARAFE instead of the traditional nearest-neighbor interpolation method, the model achieves efficient feature reorganization and expansion, improving the detection efficiency of eggplant leaf disease features, while enhancing target localization and feature representation in complex backgrounds. Lastly, by incorporating the lightweight C3-BiPC module into the neck network, the detection efficiency of small targets is effectively increased. The C3-BiPC module optimizes channel and dimension grouping through its unique design, allowing for more detailed eggplant leaf disease information to be retained, significantly enhancing the ability to extract dense small-target disease features in complex scenarios. This step not only optimizes the model’s detection performance but also ensures lightweight design while effectively improving the precise detection and processing capability for small targets, highlighting its central role and critical importance in the model optimization process. These improvements enhance the model’s ability to extract eggplant leaf disease features and provide stronger information representation, while effectively reducing the number of parameters and computational complexity, significantly enhancing the model’s performance and adaptability in complex scenarios. The following is a detailed improvement process.

#### 2.3.1. Model Lightweight

Classical residual blocks [27] are the basic building blocks for designing convolutional neural network structures. The classical residual block structure is shown in Figure 8a. Firstly, dimension reduction is carried out on the input feature graph through dimensionality reduction convolution and then dimension recovery is carried out through dimensionality enhancement convolution. These processed feature graphs are added to the original input feature graph on the residual path element by element to form the final output. Because the classical residual block contains a large number of parameters and requires a large amount of computation, it is a great challenge to deploy convolutional neural networks on embedded devices, so it is not suitable for constructing lightweight disease detection methods for eggplant leaves. The inverted residual block proposed in MobileNetv2 [28] has fewer input channels, fewer parameters, and less computation due to the use of depth separable convolution. The inverted residual block structure is shown in Figure 8b, where Dwise represents deep convolution. However, the identity mapping in inverted residuals blocks is built between low-dimensional features, so the feature compression is prone to information loss and the lower feature dimension that leads to the backpropagation is prone to gradient confusion. MobileNeXt is composed of stacked SandGlass blocks (SGBlocks), which effectively solves the problem of information loss and gradient confusion caused by low-dimensional features. Its structure is shown in Figure 8c. The channel dimension adjustment sequence of the classic residual block is followed and the bottleneck structure is moved to the middle of the hourglass block. The high-dimensional characteristics transmitted between the hourglass blocks reduce the risk of information loss and gradient confusion. At the same time, depth convolution is added at both ends of the hourglass block to extract more spatial information. This change is helpful to extract effective information of eggplant leaf disease and promote gradient flow and information transmission. This improvement in the structure of the model allows for better learning and representation of the characteristics of eggplant leaf disease in deeper layers, thereby improving the accuracy of eggplant leaf disease identification.

Mathematically, assume that the input tensor of SGBlock is
(1)$$F\in R^{D_{f}{\times D}_{f}\times M}$$

The output tensor is
(2)$$G\in R^{D_{f}{\times D}_{f}\times M}$$

The SGBlock model can be expressed as
(3)$$\hat{G}=\emptyset_{1,p}\emptyset_{1,d}(F\;)$$
(4)$$G=\emptyset_{2,d}\emptyset_{2,p}\left(\hat{G}\right)+F$$
where $\;\emptyset_{i,p}\;$ and $\emptyset_{i,d}\;$ are the i-th pointwise convolution and depthwise convolution.

Now, compare the impact of using ordinary convolution and depth-wise separable convolution on computational cost (FLOPs, floating point operations). Assume that the size of the input feature map is H×W, the number of channels is $C_{in}$, the size of the output feature map is $H^{\prime }\times W^{\prime }$, the number of channels is $C_{out}$, and the size of the convolution kernel is K×K.

Then, the FLOPs of ordinary convolution is
(5)$$FLOPs=H^{\prime }\times W^{\prime }\times C_{in}\times C_{out}\times K\times K$$

The FLOPs of depthwise separable convolution is
(6)$$FLOPs=H^{\prime }\times W^{\prime }\times C_{in}\times K\times K+H^{\prime }\times W^{\prime }\times C_{in}\times C_{out}$$

It can be seen from the above formula that when the $C_{out}\;$ and *K* values are both greater than 1, the calculation amount of depth-separable convolution is much less than that of standard convolution [29]. By improving the backbone network to MobileNeXt, it is possible to significantly reduce the complexity of the model and improve computational efficiency while maintaining network performance. This is particularly important for the task of detecting eggplant leaf diseases. It can effectively perform the task while maintaining accuracy and is particularly suitable for edge detection equipment with limited computing power. Figure 9 shows a schematic diagram of the replacement backbone network in YOLOv5s-BiPCNeXt.

During the process of upsampling, transposed convolution or bilinear interpolation is commonly employed to restore low-resolution feature maps to their original input resolution, thereby enabling a more accurate capture of target details. Nevertheless, these conventional upsampling techniques may present certain limitations. Firstly, methods such as bilinear interpolation or transposed convolution often introduce blurring and smoothing effects during traditional upsampling operations, potentially resulting in the loss of detail and target fuzziness, particularly when dealing with intricate structures like those found in eggplant leaf diseases, consequently impacting the model’s accuracy. Secondly, traditional upsampling methods may not fully exploit the information contained within low-resolution feature maps during the upsampling process, leading to potential loss of disease-related information and redundancy that restricts the model’s ability to effectively represent eggplant leaf diseases. To address these issues, the CARAFE (Content-Aware ReAssembly of FEatures) upsampling method was introduced. CARAFE is a content-aware feature reassembly method that better preserves detail and enhances feature representation during upsampling. CARAFE uses pixel-level weights to significantly improve upsampling by reorganizing information from the low-resolution feature map. It achieves this by learning dynamic weight mapping, allowing the upsampling operation to be more precise and capturing fine features of eggplant leaf diseases more effectively. The introduction of the CARAFE upsampling method effectively addresses issues of blurring, loss of detail, and information redundancy associated with traditional upsampling methods. It can further improve the accuracy and representation capabilities of the eggplant leaf disease model while maintaining the model’s lightweight characteristics.

#### 2.3.2. Optimization of the Small Target Disease Detection Model

In the previous phase of our work, the YOLOv5s-BiPCNeXt model underwent successful lightweight design optimization to meet the demands of resource-constrained environments, enabling deployment on edge detection devices while improving the operational efficiency of eggplant leaf disease identification. However, this lightweight design also introduced some challenges: reducing the model complexity and operational burden simultaneously decreased the accuracy of detecting small targets of eggplant diseases. Specifically, the original C3 module structure, although adept at handling local information, struggled to capture the global features of eggplant leaves [30]. As shown in Figure 10, the traditional bottleneck structure, primarily built using 1 × 1 and 3 × 3 standard convolutions, exhibits several notable drawbacks in this regard. When processing complex disease features, especially small targets of leaf diseases, it often becomes overly conservative, computationally intensive, and inefficient [31].

This paper proposes a new module to replace the original bottleneck structure. It includes Partial Convolution (PConv) [32], connected with two Point-wise Convolutions, and adopts BN (Batch Normalization) and ReLU activation functions. Additionally, it introduces a dynamic sparse strategy inspired by the BiFormer attention mechanism [33] to enhance and optimize the bottleneck structure. The improved C3, named C3-BiPC, is illustrated in Figure 11.

PConv is composed of subspace convolution (SSConv) and PWConv. It is an efficient and lightweight convolution module that is particularly good at extracting spatial features. Through the 3 × 3 convolution operation, the PConv module not only effectively reduces the computational cost and model parameters but also avoids frequent memory access. It can introduce more spatial feature extraction and enhance the model’s ability to handle complex disease features and small target detection. The PConv module is added to the bottleneck as a core component to replace some traditional convolution operations and utilize the redundancy of the eggplant leaf disease feature map to further optimize the cost. Due to the high similarity between different channels in the leaf feature map, the number of original convolution channels is reduced by a certain proportion using a typical ratio.

This ratio is denoted as $r=\frac{C_{P}}{C}=\frac{1}{4}$. That is, the number of channels selected for the partial convolution is $C_{p}=\frac{1}{4}C$. The formula for the FLOPs of the ordinary convolution is obtained by (5). When the input and output image sizes are the same, the formula becomes
(7)$$FLOPs=W\times H\times K^{2}\times C^{2}$$

Standard convolution is used to extract spatial features on the input channels $C_{p}$ while keeping the rest of the channels unchanged. The structure is shown in Figure 12. The first or last consecutive $C_{p}$ channels are considered as representatives of the entire feature map for continuous or regular memory access. It is assumed that the input and output feature maps have the same number of channels. Only SSConv has FLOPs.
(8)$$FLOPs=W\times H\times K^{2}\times C_{p}^{2}$$

When extracting spatial features, a strategy of selectively activating a portion of the $C_{p}$ channels is adopted, while retaining the remaining (C$-C_{p}$) channels without pruning. By keeping the remaining channels unchanged, this design allows feature information to flow through all channels in the PWConv layer. The computational effort of PConv is only the sum of the computational effort of SSConv and PWConv. At this point, the FLOPs of the structure equate as follows:(9)$$FLOPs=H\times W\times (K^{2}\times C_{p}^{2}+C^{2})$$

Compared with ordinary convolution, PConv will use smaller FLOPs to complete feature information extraction. In the bottleneck, the PConv position is added as shown in Figure 13.

In order to better understand and process the key information in eggplant leaves, a BiFormer attention mechanism is added to the output of residual structure information to enhance the fusion and representation capabilities of features.

BiFormer is a new dynamic sparse attention mechanism based on bi-level routing, aimed at achieving more flexible computational allocation and content-awareness. This approach addresses the scalability problem of multi-head self-attention by enabling dynamic query-aware sparsity. It consists of multiple BiFormer Blocks, as illustrated in Figure 14.

The key component of the BiFormer Block is the Bi-level Routing Attention (BRA) module. BRA employs a coarse-grained region-level approach to exclude the least relevant key-value pairs, instead of operating at the fine-grained token level. This strategy creates a directed graph and computes a measure of inter-region connectivity to detect the most relevant k regions. Finally, using the region-based routing index matrix, fine-grained token-to-token attention is achieved. By collecting key-value pairs and applying attention mechanisms to them, along with a local context enhancement function, BRA efficiently handles the routing regions spread throughout the feature map, thereby optimizing computational performance. The working principle of BRA is shown in Figure 15. Enter an image, $X\in R^{H\times W\times C}$, and first divide it into the S×S distinct regions, where each region contains $\;\frac{HW}{S^{2}}$ feature vectors.

The two-dimensional input feature graph $X\in R^{H\times W\times C}$ is divided into S x S nonoverlapping regions. Each region contains $\frac{HW}{S^{2}}$ feature vectors and X is remapped to $X^{r}\in R^{S^{2}\times \frac{HW}{S^{2}}\times C}$. We then derive the query, key, value tensor, Q, K, and V, with linear projections, as follows:(10)$$Q=X^{r}W^{q}\;\;,K=X^{r}W^{k}\;\;,V=X^{r}W^{v}$$
where $W^{q}$, $W^{k}$, and $W^{v}$ are projection weights for the query, key, and value, respectively.

Construct a directed graph to represent the attention relationships between different areas in the input image.

The query matrix $Q^{r}$ and key matrix $K^{r}$ are obtained by averaging the queries and keys of each region. Then, the adjacency matrix $A^{r}$ describing the relation between regions obtained by matrix multiplication of $Q^{r}$ and transposed $K^{r}$.
(11)$$A^{r}=Q^{r}{\left(K^{r}\right)}^{T}$$

This adjacency matrix $A^{r}$ represents the strength of the association between each region and other regions and is used to construct the attention relationship diagram between regions.

This affinity graph is then pruned to keep only the first k connections that are most relevant for each region. This is achieved by the row-wise topk operator, that is, for each row (representing a region), only the k connections with the highest degree of correlation with other regions are retained, resulting in a routing index matrix $I^{r}$.
(12)$$I^{r}=topkIndex\;(A^{r})$$

Therefore, the $I^{r}$ row of the matrix i contains k feature map regions of the region most relevant to the i region. This process realizes the information exchange between different target tasks, successfully filters out the information that is highly relevant to the detection target, and makes these information dominate in the model learning process, while other irrelevant information is suppressed, so as to effectively allocate attention.

Based on this index matrix $I^{r}$, one can further obtain the tensor $K^{g}$, $V^{g}$ of the keys, and values collected by each query in the region i.
(13)$$K^{g}=gather\;(K,I^{r})\;\;,V^{g}=gather\;(V,I^{r})$$
where $K^{g}$ and $V^{g}$ are the gathered key and value tensor.

Then, apply attention on the gathered key-value pairs as
(14)$$O=Attention\;(Q,K^{g},V^{g})+\mathrm{LCE}\;(V)$$

Attention in the formula represents the regular self-attention mechanism, while LCE(V) is a local context enhancement term, parameterized with deep convolution.

After the key-value pair is collected, the attention mechanism is used to obtain the final optimized result O.

Overall, the introduction of the C3-BiPC module has a significant positive impact on the eggplant leaf disease detection model. These improvements enhance the model’s ability to understand and handle complex and rich features, particularly under limited computational capacity, while maintaining good performance. Not only does this boost the model’s ability to parse and discern local and global features, as well as cross-scale and shape-related disease information, but it also increases the model’s focus on important features, further enhancing the detection accuracy of eggplant leaf diseases. Specifically, the inclusion of Partial Convolution (PConv) enables selective activation of certain channel features, allowing the model to exhibit complex and rich nonlinear characteristics even under lower computational capacity. On the other hand, with the aid of the BiFormer attention mechanism, the model’s capacity for processing key information is strengthened, demonstrating excellent small-object disease feature detection ability while effectively understanding and managing relationships among various diseases. Additionally, BiFormer further improves the model’s computational efficiency, especially at the coarse-grained region level, by filtering key-value pairs for efficient computation.

The C3-BiPC module enables the eggplant leaf disease detection model to have better accuracy in recognition and processing while maintaining efficient calculations, especially the improvement of small target recognition performance, allowing the eggplant leaf disease detection model to maintain efficient calculations while maintaining efficient calculations, with higher recognition and processing accuracy, thus significantly enhancing its practical application value in agriculture, including real-time monitoring of small lesions, precise diagnosis, optimized pesticide use, improved yield and quality, and supporting data-driven agricultural decision making.

#### 2.3.3. Optimization of Disease Target Detection at Different Scales

In order to inflow the effective information of eggplant leaf disease characteristics into the feature fusion network before the C3-BiPC module that learns the residual features, an efficient multi-scale EMA attention mechanism module is inserted and the feature information is screened once. By integrating feature information containing more eggplant disease characteristics into the neck network, the global contextual connections are further strengthened, enhancing the model’s detection capability. This approach addresses the issue of target loss or misjudgment in cases of overlap, while optimizing detection performance for targets of different scales, as shown in Figure 16.

In this module, a variety of operations are used to enhance feature extraction and representation capabilities. First, the input feature graph is divided into several groups in terms of channel dimension by group convolution to learn different semantic information. Then, 1D global average pooling (X avg Pool) in the horizontal direction and 1D global average pooling (Y avg Pool) in the vertical direction were carried out to capture the global information in the corresponding direction. Then, the group 3 × 3 convolution (Groups (3 × 3)) is applied and the feature map after the group is checked by 3 × 3 convolution for convolution operation to capture multi-scale features. Then, multiple feature maps are spliced together (Concat) and feature fusion is performed by 1 × 1 convolution (1 × 1). To improve the performance of the model, the Sigmoid activation function is used to compress the input values to between 0 and 1, generating probability values or weights. At the same time, the re-weight operation is used to adjust the importance of different channels by calculating the attention weight, so as to enhance the feature representation of the region of interest. Finally, the feature graph is reduced in size by global average pooling (Avg Pool) while preserving important global information, and the input vector is converted into a probability distribution by Softmax activation function, which is often used in the output layer of multi-classification problems. In addition, Groupnorm technology divides feature maps into groups and normalizes them within each group to speed up training and improve stability. Matrix multiplication (Matmul) operations enhance feature representation by multiplying two matrices together to combine features.

The EMA attention mechanism uses three parallel paths to extract attention weights for grouped feature maps, optimizing high-level feature maps for eggplant leaves without reducing the channel dimension, achieving better pixel-level attention. The 3 × 3 branch processes the feature map through 3 × 3 convolution operations, effectively capturing cross-dimensional interactions and establishing links between different dimensions with other branches. Meanwhile, the output from the X and Y branches undergoes 2D global average pooling to encode global spatial information. Before jointly activating the channel features, the output from the smallest branch is converted to the corresponding dimensional shape, further facilitating cross-spatial information gathering. After the EMA attention mechanism extracts feature information from the eggplant leaves, it calculates the attention weights for each pixel. Based on the feature differences between diseased and healthy leaves, the EMA mechanism extracts pixel information through parallel X and Y branch paths and determines the weights for these features. Then, two spatial attention maps are generated from the calculated weights, highlighting key differences between diseased and healthy areas at the image scale. The model weights the original feature map based on these two spatial attention maps, enhancing the significance and prominence of disease-related features in the model, while appropriately suppressing or ignoring less important features (such as background or non-disease portions). The next step is to nonlinearly transform the adjusted feature map and restore it to the same size as the input feature map, enabling further processing by subsequent modules. In this way, the feature map processed by the EMA attention mechanism is rich in disease-related information, aiding in more accurate detection of eggplant leaf diseases by emphasizing critical regions and disregarding irrelevant features. Moreover, cross-spatial attention feature extraction significantly improves the detection of eggplant leaf diseases of different scales. Ultimately, the result is a model that, after EMA attention mechanism processing, achieves higher accuracy in distinguishing eggplant leaf disease features.

## 3. Experimental and Results Analysis

### 3.1. Experimental Setting

In the experimental setup, a series of fine-tuning adjustments are made to optimize the model’s performance. Specifically, the following settings are configured: the number of training epochs is set to 200, with a batch size of 16. The initial learning rate is set at 0.01, accompanied by a learning rate momentum of 0.937 [34]. Stochastic Gradient Descent (SGD) is selected as the optimization strategy and the input image resolution is standardized to 640 × 640. Additionally, the weight decay coefficient is set to 0.0005. Beyond these configurations, all other settings are optimized for the best experimental results. Other experimental environment details are presented in Table 2.

### 3.2. Evaluation Criteria

In this study, the model is comprehensively evaluated using various metrics, including precision (P), recall (R), average precision (AP), mean average precision (mAP), model weights (MB), giga floating-point operations (GFLOPs), and parameters and frames per second (FPS). These metrics offer a comprehensive assessment of the model’s performance. Precision is the proportion of correctly predicted positive samples to the total number of predicted positive samples, which is used to measure the classification ability of the model, while the recall rate measures the proportion of correctly predicted positive samples to the total number of positive samples.

AP is the integral of precision and recall and mAP is the average of AP, reflecting the overall performance of the model for target detection and classification. The F1 score is the harmonic mean of precision and recall, which uses both precision and recall to evaluate the model performance. The specific calculation formula is shown as follows.
(15)$$Precision=\frac{TP}{TP+FP}$$
(16)$$Recall=\frac{TP}{TP+FN}$$
where TP is the number of positive samples for correct detection, FP is the number of positive samples for error detection, and FN is the number of negative samples for error detection.
(17)$$AP=\int_{0}^{1}P(R)dR$$
(18)$$mAP=\frac{\sum_{i=1}^{n}AP_{i}}{n}$$
where n is the number of disease species.

Model weight refers to the amount of memory required to store the model. GFLOPs is a measure of the model’s complexity, representing the total number of multiplication and addition operations the model performs. The lower the GFLOPs value, the less computational power is needed for model inference, leading to faster model computation. $\;{mAP@}_{0.5}\;$ indicates the *m**A**P* value when IoU is 0.5, ${mAP@}_{0.5:0.95}$ indicates the mAP value when IoU is in the interval from 0.5 to 0.95, and the interval value of IoU is 0.05.

### 3.3. Detection Performance of YOLOv5s-BiPCNeXt

Under identical experimental conditions, including the same dataset, uniform training parameters, and identical hardware environment, the analysis of the YOLOv5s-BiPCNeXt model’s performance on a custom eggplant leaf dataset, compared with the YOLOv5s model, is shown in Table 3. The results indicate that the YOLOv5s-BiPCNeXt model achieved a P, R, and mAP of 96.2%, 94.8%, and 96.5%, respectively. These represent increases of 0.3%, 1.5%, and 1.2% compared to YOLOv5s. Additionally, the model’s weight is only 8.8 MB, a reduction of 35% compared to the original YOLOv5s.

The curve comparison diagram of $\;{mAP@}_{0.5}\;$ and ${\;mAP@}_{0.5:0.95}$ of YOLOv5s-BiPCNeXt and YOLOv5s is shown in Figure 17. It can be seen that in 200 training, ${mAP@}_{0.5}$ and ${\;mAP@}_{0.5:0.95}$ of YOLOv5s-BiPCNeXt are always consistently higher than YOLOv5s.

The confusion matrix is characterized by a row vector representing the prediction class and a column vector representing the actual class, as shown in Figure 18. It includes three major categories of eggplant leaves: healthy leaves, brown spot leaf leaves, and powdery mildew leaf leaves. We carefully analyzed the accuracy of predicting various leaf classes and it is clear that the improved model has a significant increase in recall rates compared to the original model. Specifically, recall rates for healthy leaves, brown spots, and mold were 1.00, 0.95, and 0.95, respectively, and most of the leaf disease categories were correctly detected, indicating that the improved model enhanced image feature extraction capabilities and thus improved detection accuracy.

### 3.4. Ablation Study

To validate the effectiveness of the YOLOv5s-BiPCNeXt model improvements, an ablation study is designed based on the YOLOv5s-BiPCNeXt, with all experimental parameters held constant, to analyze the impact of different optimization modules on model performance. The study involves introducing various optimization modules individually into the YOLOv5s baseline model, including the MobileNeXt backbone network, CARAFE upsampling technology, the addition of the C3-BiPC module, and the integration of the EMA attention mechanism. The experimental results are presented in Table 4.

To be more specific, YOLOv5s had a ${mAP@}_{0.5}$ of 94.3%. By replacing the backbone with the MobileNeXt architecture, the model’s parameter count decreased by 45.6%, achieving a significant reduction. However, this resulted in a slight loss of precision, with a decrease of 2.8% compared to YOLOv5s. When the C3-BiPC module was added on top of this, the saw a significant increase, reaching 96.0%. This improvement reduced the FLOPs while enhancing the effectiveness of feature information aggregation. Replacing the nearest-neighbor interpolation with the CARAFE upsampling method resulted in a slight ${mAP@}_{0.5}$ increase of 0.2%, while reducing the parameter count, indicating the effectiveness of CARAFE in context aggregation and content-aware processing. In summary, by using MobileNeXt as the backbone network and changing the upsampling strategy to CARAFE, the model successfully reduced the parameter count and computational complexity, albeit with a slight decrease in precision. To mitigate this, the introduction of the C3-BiPC module enhanced the model’s ability to understand and process complex and rich features, improving the detection precision for small-object disease characteristics. Furthermore, integrating the EMA attention mechanism effectively enhanced the network’s multi-dimensional information perception, optimizing the detection of leaf diseases across different scales. As a result, the YOLOv5s-BiPCNeXt model achieved optimal performance through these adjustments and improvements.

### 3.5. Contrasting Experimental Studies of Multiple Attention Mechanisms

To validate the effectiveness of the EMA attention mechanism module, typical attention mechanism modules such as CA (Coordinate Attention), SE (Squeeze-and-Excitation), and CBAM (Convolutional Block Attention Module) were used for performance comparison, as shown in Table 5. The results demonstrate that introducing an attention mechanism significantly enhances model accuracy. Specifically, models with CA, SE, CBAM, and EMA had ${mAP@}_{0.5}\;$ scores of 94.5%, 93.2%, 95.0%, and 96.5%, respectively, representing improvements of 0.2%, 0.5%, 0.7%, and 2.2% compared to YOLOv5s. Among these comparisons, the EMA-based model performed best, achieving an ${mAP@}_{0.5}$ of 96.5%, surpassing CA, SE, and CBAM by 2%, 1.8%, and 1.5%, respectively. This could be due to the EMA attention mechanism’s ability to combine both spatial and channel attention dimensions, highlighting disease-related information on the feature map more distinctly, thus improving the model’s disease detection and localization capabilities. This evidence supports the important role of the EMA attention mechanism in enhancing model performance, especially in scenarios like disease detection, where it has significant advantages.

### 3.6. Comparative Performance Analysis of the Different Models

To further validate the superiority of YOLOv5s-BiPCNeXt, a comparison evaluation was conducted with Faster R-CNN, YOLOv5s, YOLOv7, YOLOv8s, YOLOv9-C, and YOLOv5s-BiPCNeXt, maintaining all experimental parameters constant. The experimental results are presented in Table 6.

The results indicate that among all the involved models, the two-stage detection model Faster R-CNN presented a relatively high $\;{mAP@}_{0.5}$ of 95.8%, slightly exceeding YOLOv5s, and only 0.7% lower than YOLOv5-BiPCNeXt. However, the Faster R-CNN model has a large model weight and high GFLOPs, leading to a detection speed of only 8 FPS, which is not suitable for eggplant leaf disease detection tasks requiring high real-time performance. YOLOv7, YOLOv8s, and YOLOv9-C all have high parameter counts and model weight sizes, with the relatively new YOLOv9-C model having a parameter count of 51,004,178, resulting in a slightly higher ${mAP@}_{0.5}$ of 97.2% but at the cost of requiring high computing power, which imposes certain requirements on the devices where the model is deployed, making it unsuitable for real-time eggplant leaf detection tasks. In contrast, YOLOv5s-BiPCNeXt achieves an ${mAP@}_{0.5}$ of 96.5%, exceeding YOLOv5, YOLOv7, and YOLOv8 by 2.2%, 0.1%, and 2%, respectively. Additionally, it significantly meets the need for model lightweighting, with a parameter count of 4,151,808, GFLOPs of only 9.8, and a model weight of 8.8 MB. Due to its lower computational complexity, the YOLOv5s-BiPCNeXt model achieves a detection speed of 26 FPS, sufficient to meet real-time detection requirements in practical applications for eggplant leaf diseases. Overall, the YOLOv5s-BiPCNeXt model demonstrated excellent overall performance, enabling the accurate and efficient detection of eggplant leaf diseases in real-world applications. The radar chart in Figure 19 provides a clear comparison of various parameter metrics across different models.

### 3.7. Comparison of Blade Detection Capabilities of Different Models in Real Scenarios

To comprehensively evaluate and compare the capability of various advanced models to detect eggplant leaf diseases in real-world scenarios, six models were tested in this experiment: Fast R-CNN, YOLOv5s, YOLOv7, YOLOv8s, YOLOv9-C, and the YOLOv5s-BiPCNeXt model constructed in this study. Each model was deployed on edge detection devices, as shown in Figure 20. The edge detection devices included the Jetson Orin Nano development board, combined with the D435i depth camera, used to detect images of affected eggplant leaves to evaluate the disease detection performance of each model. Through this approach, the effectiveness of each model was empirically studied using a self-established eggplant leaf disease image dataset.

Figure 21 shows the detection performance of each model on edge detection devices for different leaves, with powdery mildew marked in red boxes, brown spots in blue boxes, and healthy leaves in pink boxes. In each set of photos, ‘a’ represents a healthy leaf, ‘b’ represents a leaf with brown spots, ‘c’ represents a leaf with powdery mildew, and ‘d’ represents a leaf with both brown spots and powdery mildew. The experimental results demonstrate varying performance among different models in detecting eggplant leaf diseases. First, compared to other models, YOLOv5s’s overall detection accuracy did not meet expectations, especially in small-target detection. For instance, in the first image ‘a’, its detection accuracy for small healthy leaves was only 88.0%, and in the fourth detection image ‘d’, the model failed to detect a small patch of brown spot on eggplant leaves. YOLOv7 showed improvement compared to YOLOv5s, with more than 90% accuracy in detecting small brown spots. However, its accuracy and precision still had room for improvement. Looking at YOLOv8s, although it achieved 98.0% accuracy in detecting healthy leaves, the model was still weak at detecting smaller diseased targets. For example, when detecting powdery mildew, it showed overlapping bounding boxes, indicating a need for further optimization in bounding box localization quality, especially in crowded object scenarios. Fast R-CNN and YOLOv9-C exhibited improved accuracy, with YOLOv9-C reaching 99% accuracy in detecting healthy leaves. However, these models had limitations in detecting small targets and distinguishing overlapping targets. Both models occasionally missed small brown spots. YOLOv5s-BiPCNeXt, on the other hand, showed significant advantages. It achieved 99% accuracy in detecting healthy leaves and could accurately mark brown spots on various scales. This model’s accuracy in detecting small targets reached 91% or higher. It successfully detected leaves with two types of diseases and showed no overlapping bounding boxes, greatly reducing the likelihood of overlooking small diseased targets. Overall, YOLOv5s and YOLOv8 had relatively lower accuracy. YOLOv7, Fast R-CNN, and YOLOv9-C improved but they still faced challenges in detecting small disease targets. YOLOv8 and Fast R-CNN struggled with overlapping targets, potentially limiting their applications in complex detection scenarios. In contrast, YOLOv5s-BiPCNeXt displayed significant advantages in detecting leaf diseases, with strong detection capabilities for various scales and small targets. Moreover, it could successfully detect overlapping and obscured disease symptoms. This enhanced detection capability not only demonstrates the model’s high usability and robustness in real-world plant cultivation scenarios but also highlights its great potential in real-time plant disease detection.

In order to further verify the improvement in YOLOv5s-BiPCNext’s performance in detecting eggplant leaf disease, we selected challenging tiny eggplant leaf disease images from the dataset, set the original light and dark light images for testing, and compared the performance of YOLOv5s-BiPCNext, YOLOv5s, and the higher accuracy YOLOv9-C model. We analyzed the confidence levels and randomly selected detection results of each disease, as shown in Figure 22. The results demonstrate that YOLOv5s-BiPCNext outperforms the other models.

The detection results under different interference conditions show that under normal illumination, the recognition accuracy of YOLOv5s-BiPCNeXT is very different from that of YOLOv9-C and the overall accuracy of YOLOv5s is relatively low. In the dark light condition, the original features of some images will be changed and the features extracted by the convolutive neural network contain a large amount of interference information. When detecting brown spots and powdery mildew, YOLOv9-C will fail to detect some small targets, while YOLOv5s will also fail to detect the diseases of small targets and the overall accuracy will also decrease. YOLOv5s-BiPCNeXt diseases can be correctly identified basically and the impact on detection accuracy is very low.

## 4. Discussions

### 4.1. Lightweight Model Optimization

In the field of agricultural detection, lightweight model optimization has been a crucial research direction. Jiuxin Wang [35]. utilized the inverted residual convolution modules of MobileNetv2, replacing the standard convolution modules of the YOLOv5 backbone to detect apples. This approach successfully reduced the model size by 57% and improved data processing speed by 26.8%. However, the structure of inverted residual blocks may lead to information loss and the lower feature dimensionality can exacerbate gradient confusion issues. Additionally, due to the larger size of mature apple targets, the challenge of recognizing small targets was not addressed by the authors. In this study, we employed a lightweight model based on the MobileNeXt backbone, utilizing stacked Hourglass blocks (SGBlocks) to effectively mitigate information loss and gradient confusion caused by low-dimensional features. This significantly enhanced inference speed while reducing network parameters and computational complexity almost without sacrificing accuracy. Furthermore, the introduction of the Content-Aware Reassembly of Features (CARAFE) operator, replacing traditional nearest-neighbor interpolation, not only reduced computational demand but also notably enhanced the spatial resolution of disease feature maps.

### 4.2. Accuracy Enhancement

Our approach strikes a balance between lightweight optimization and accuracy, contrasting sharply with the existing literature on lightweight techniques. For instance, Ange Lu [36] employed GSConv and VoVGSCSP modules to reduce model parameters and volume without deeply addressing the trade-off between inference speed and accuracy. Similarly, Cui [37] achieved certain effectiveness in pine cone detection using squeeze-and-excitation feature pyramid networks but did not significantly improve inference speed and spatial resolution. This study introduces a multiscale spatial-cross attention mechanism (EMA) to enhance the perception of multidimensional information and extract multiscale disease features. The specially designed lightweight cervical optimization module C3-BiPC effectively reduces redundant computation and memory usage, optimizing global feature extraction of eggplant leaves. The application of dynamic query-aware models further enhances the capture capability of small-sized disease targets, as evidenced in ablation experiments where the introduction of C3-BiPC alone significantly improves detection accuracy in yolov5s. Moreover, our method thoroughly compares with existing research in the introduction of attention mechanisms. At the same time, our method also makes an in-depth comparison with existing research on the introduction and optimization of attention mechanisms. For example, Feng Xiao et al. [38] integrated the CBAM attention mechanism in YOLOv5 to improve the accuracy of blueberry fruit detection but did not discuss its application effect in multi-scale and small target detection in detail. Our study significantly enhanced the feature extraction capability of the YOLOv5s-BiPCNeXt model through the EMA attention mechanism and verified its superiority in recognition accuracy over CA, SE, CBAM, and other attention mechanisms in the experiment. This may be due to the fact that other attention mechanisms, such as SENet [39] and CBAM [40], only consider inter-channel information or local relationships, while the EMA attention mechanism can combine spatial and channel attention dimensions to highlight disease-related information more clearly on the feature map, thereby improving the model’s disease detection and location capabilities, which is important for capturing the target structure in the detection task.

### 4.3. Future Outlook

The proposed YOLOv5s-BiPCNeXt model demonstrates promising detection performance in limited sample testing, particularly excelling in low-light conditions compared to other algorithms, reducing instances of missed and false detections while significantly improving accuracy. Nonetheless, our study still faces challenges in further enhancing the model’s adaptability to various adverse weather conditions. Future work will focus on expanding the dataset to include a wider range of plant leaf disease images and exploring the model’s performance on edge computing devices such as Jetson Orin Nano, integrated with spraying devices, to achieve automated detection and management of eggplant leaf diseases.

## 5. Conclusions

In this study, we proposed the YOLOv5s-BiPCNeXt lightweight detection model to effectively detect eggplant leaf diseases. The MobileNeXt backbone network was incorporated to significantly reduce the model’s parameter count and computational complexity. Specifically, the C3-BiPC neck module was designed to dynamically query and capture sparse information, enhancing the detection accuracy of small disease spots. The multi-scale cross-space attention mechanism (EMA) and the content-aware feature recombination operator (CARAFE) were integrated into the neck network. These mechanisms improved the model’s ability to perceive multi-dimensional information and extract multi-scale disease features, thereby enhancing the spatial resolution of the disease feature map. Experimental results demonstrated that the YOLOv5s-BiPCNeXt model achieved an mAP@0.5 of 96.5%. Compared to the original YOLOv5s model, the YOLOv5s-BiPCNeXt reduced the number of parameters by 41%, computational cost by 37.9%, and model weight by 35.2%, resulting in 4,151,808 parameters, 9.8 GFLOPs, and 8.8 MB, respectively. The model achieved an average detection speed of 26 FPS, meeting the real-time detection requirements. The detection AP for brown spot and powdery mildew were 94.9% and 95.0%, respectively, with an AP of 99.5% for healthy leaves. These results indicate that the YOLOv5s-BiPCNeXt model excels in detection accuracy and real-time performance, making it highly suitable for deployment on resource-constrained edge detection devices. Future research could focus on deploying the YOLOv5s-BiPCNeXt model on edge devices for real-time leaf disease detection tasks, potentially integrating with robotic sprayers or real-time monitoring systems to enhance agricultural productivity and address eggplant diseases promptly.

## Figures and Tables

**Figure 1 plants-13-02303-f001:**
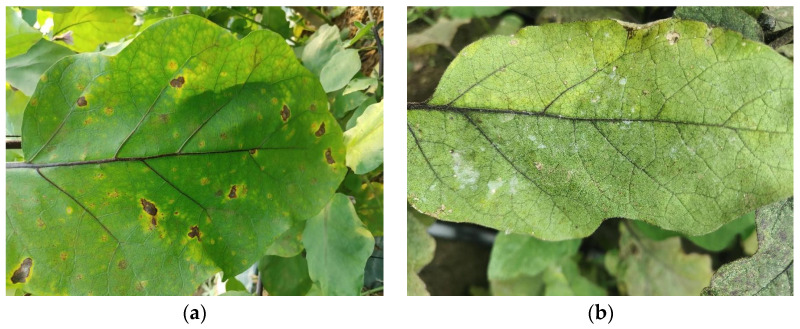
Eggplant leaf diseases: (**a**) leaves with brown spot disease and (**b**) leaves with powdery mildew.

**Figure 2 plants-13-02303-f002:**
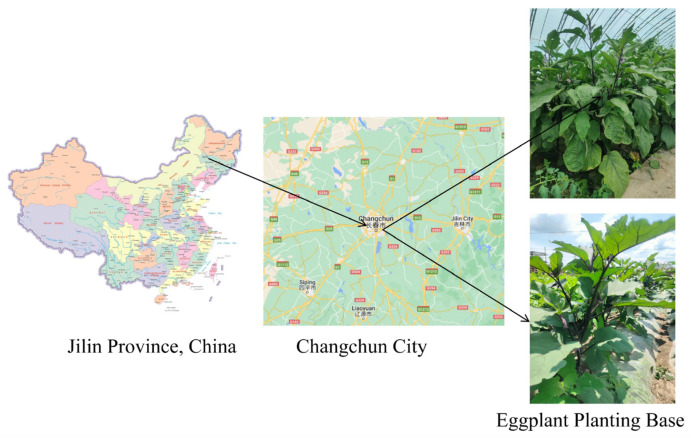
Geographic location of the data acquisition.

**Figure 3 plants-13-02303-f003:**
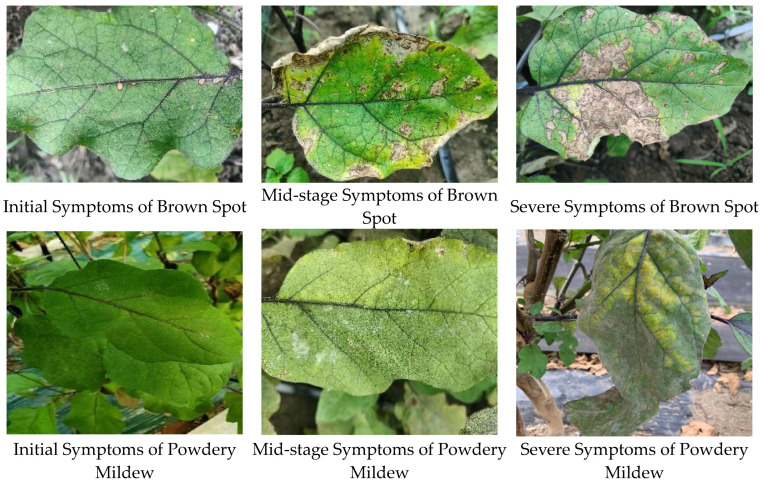
Diseased leaves of varying severity.

**Figure 4 plants-13-02303-f004:**
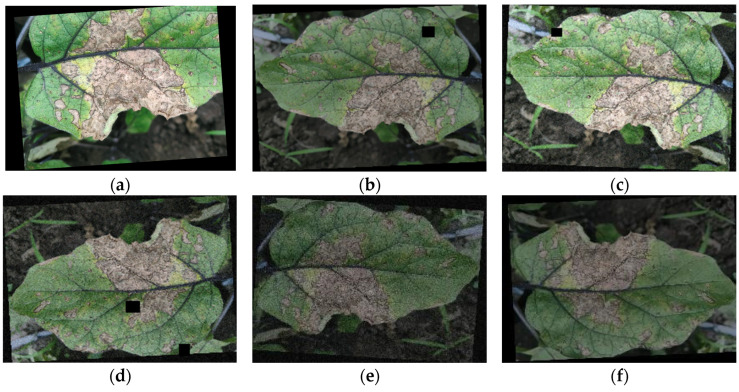
Different data enhancements are performed on the images to expand the dataset: (**a**) image amplification + rotation; (**b**) random occlusion + flip horizontal+ adjusted brightness; (**c**) flip horizontal+ random occlusion; (**d**) rotation + random occlusion+ adjusted brightness; (**e**) adjusted brightness+ rotation; and (**f**) flip horizontal + adjusted brightness.

**Figure 5 plants-13-02303-f005:**
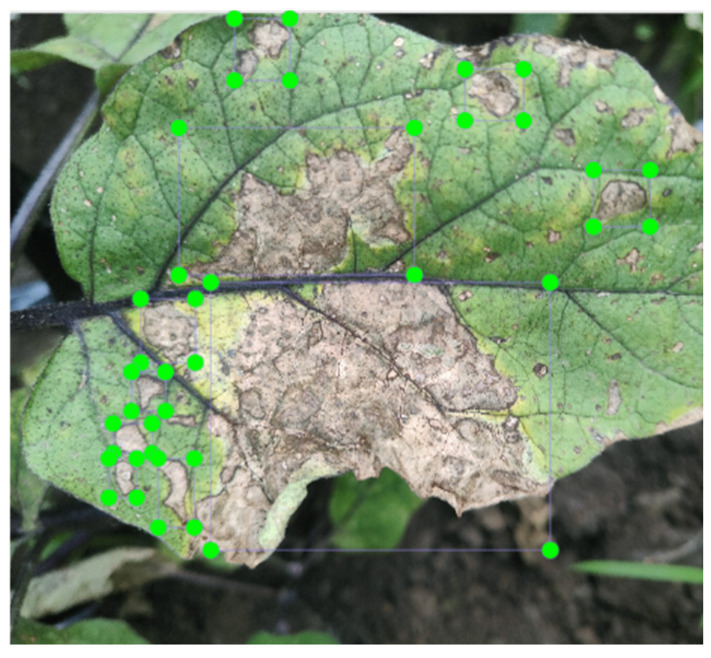
Diseased areas of eggplant leaves were marked using LabelImg software.

**Figure 6 plants-13-02303-f006:**
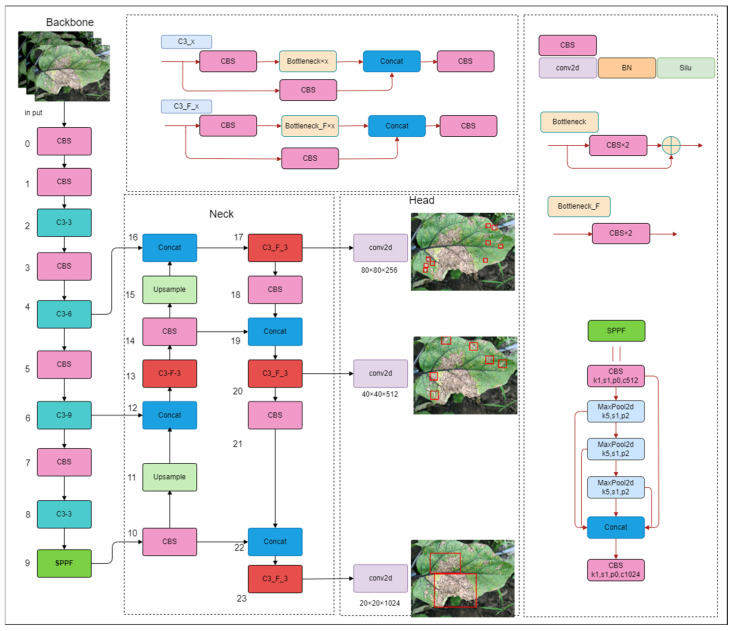
YOLOv5s structure diagram.

**Figure 7 plants-13-02303-f007:**
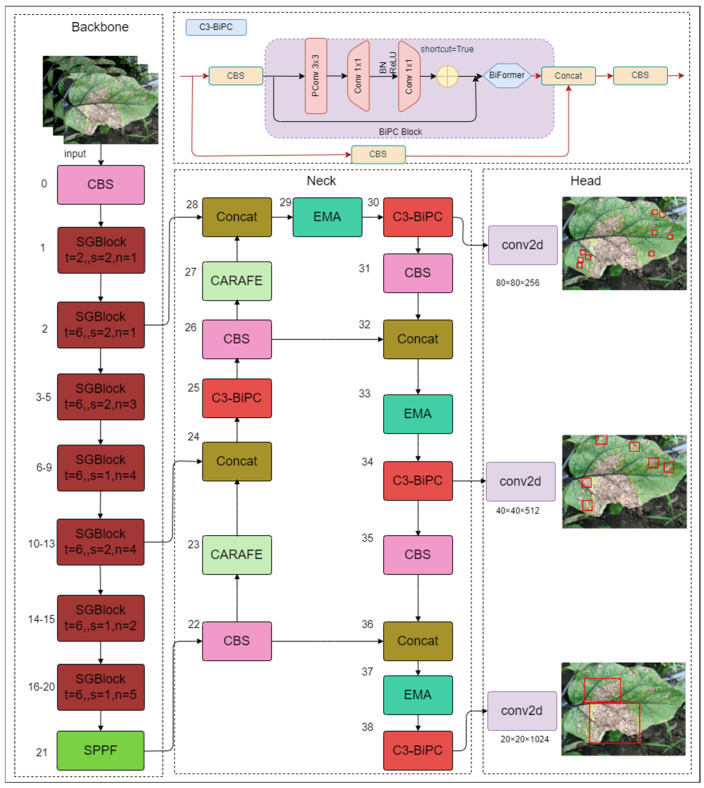
Structure of YOLOv5s-BiPCNeXt.

**Figure 8 plants-13-02303-f008:**
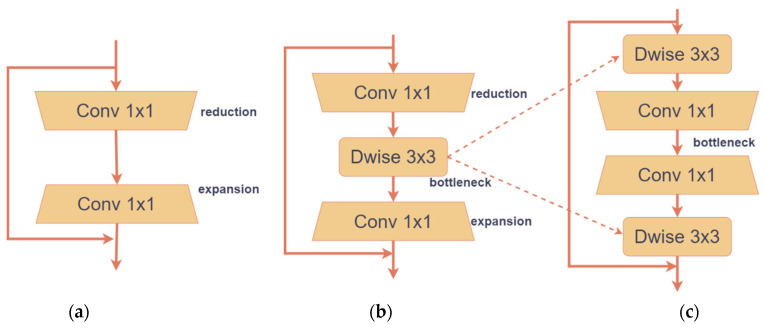
Different residual modules: (**a**) classic residual block; (**b**) inverted residual block; and (**c**) sandglass block.

**Figure 9 plants-13-02303-f009:**
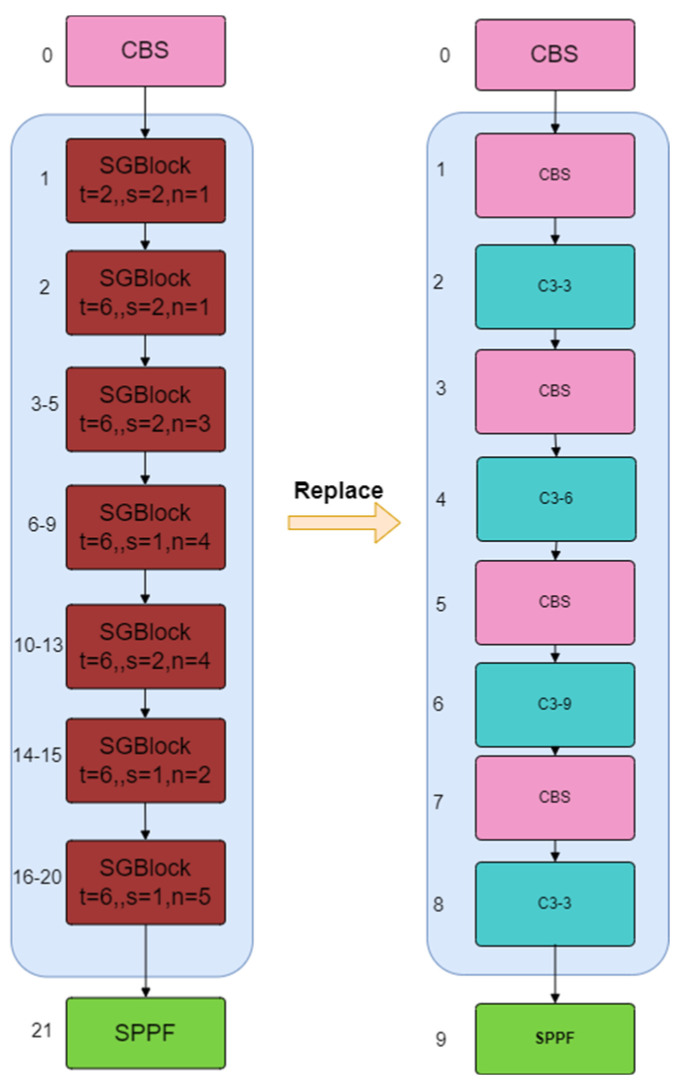
Replace the backbone network of YOLOv5s with SGBlock in MobileNeXt.

**Figure 10 plants-13-02303-f010:**
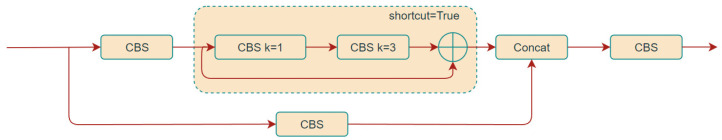
The C3 module.

**Figure 11 plants-13-02303-f011:**
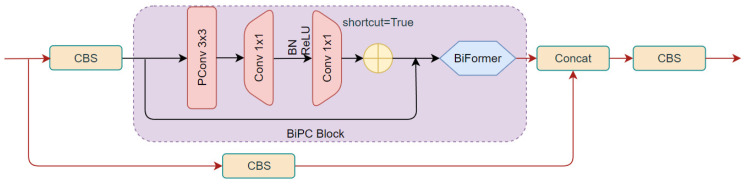
The C3-BiPC module.

**Figure 12 plants-13-02303-f012:**
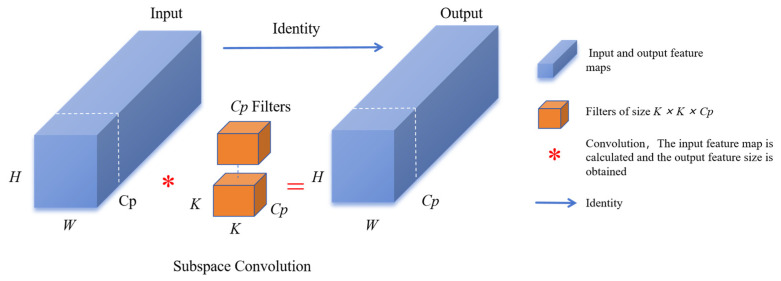
Subspace convolution structure diagram.

**Figure 13 plants-13-02303-f013:**
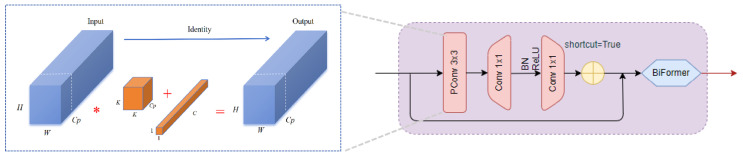
Improved bottleneck.

**Figure 14 plants-13-02303-f014:**
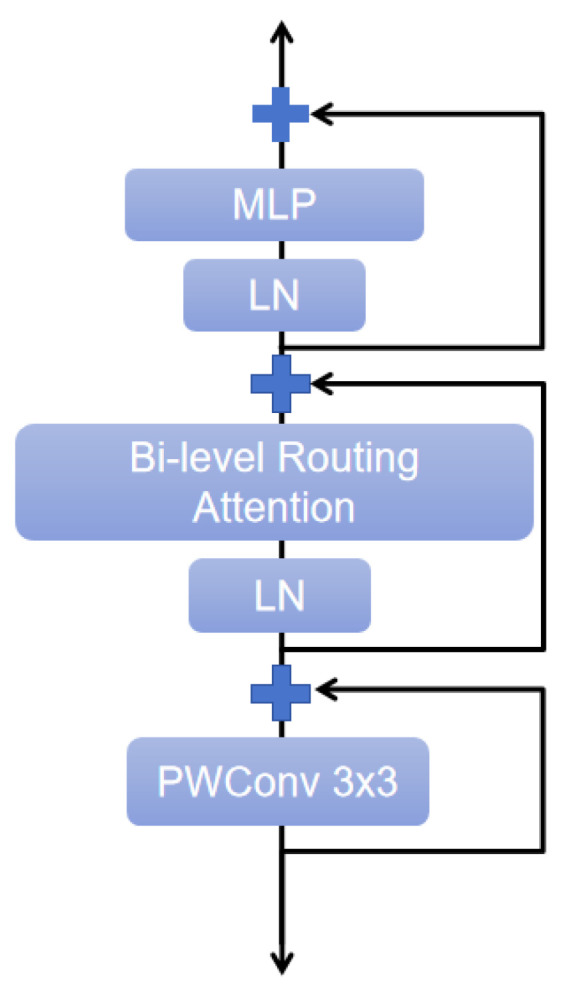
BiFormer Block.

**Figure 15 plants-13-02303-f015:**
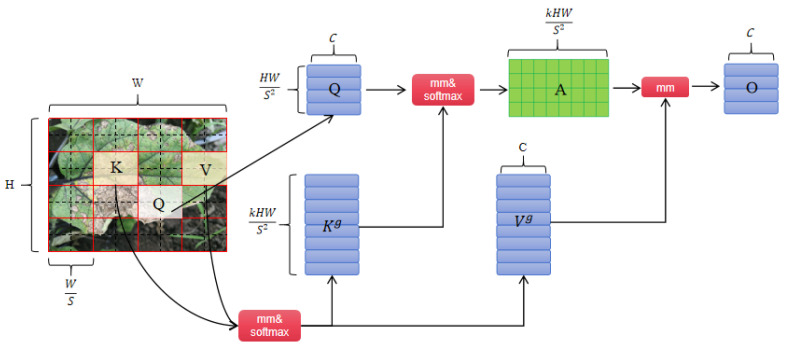
BRA module structure diagram.

**Figure 16 plants-13-02303-f016:**
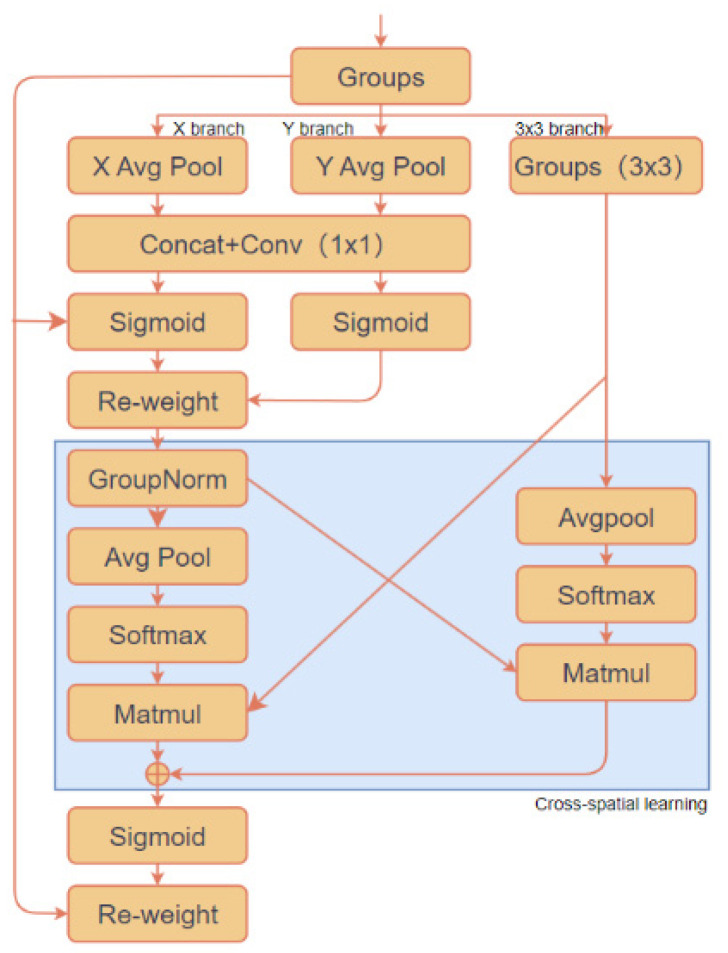
Efficient multi-scale attention module.

**Figure 17 plants-13-02303-f017:**
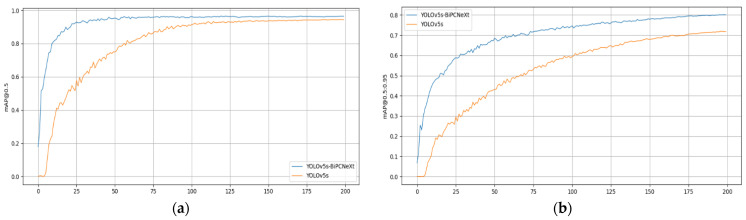
Precision comparison curve of YOLOv5s and YOLOv5s-BiPCNeXt at epoch of 200: (**a**) ${mAP@}_{0.5}$ comparison curves of the two models and (**b**) ${mAP@}_{0.5:0.95}$ comparison curves of the two models.

**Figure 18 plants-13-02303-f018:**
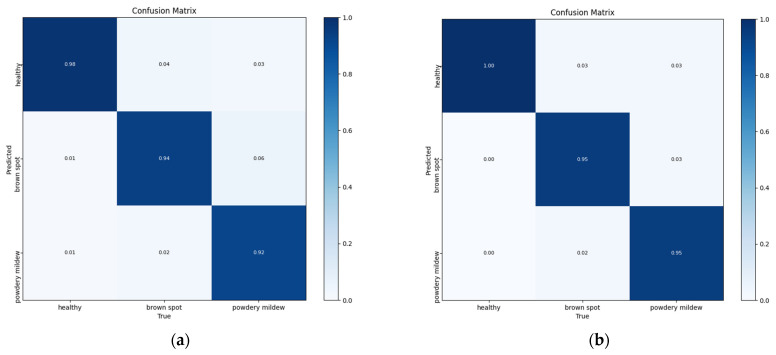
YOLOv5s (**a**) and YOLOv5-BiPCNeXt (**b**) confusion matrix diagram.

**Figure 19 plants-13-02303-f019:**
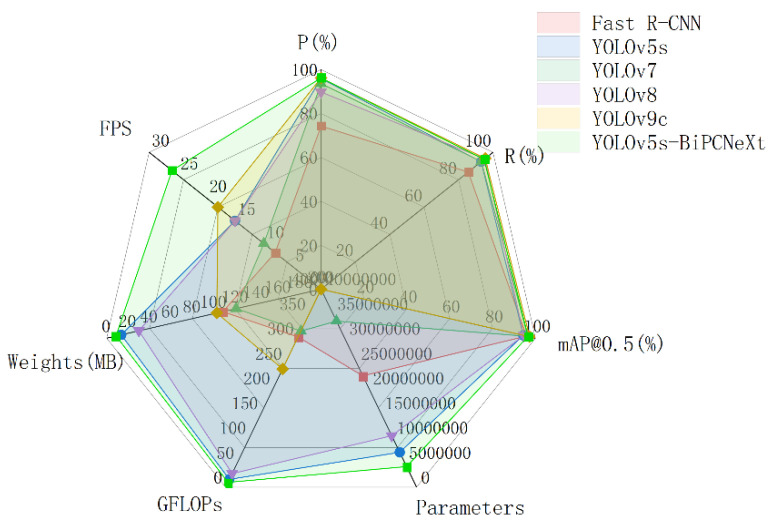
Performance comparison radar chart of different detection algorithms.

**Figure 20 plants-13-02303-f020:**
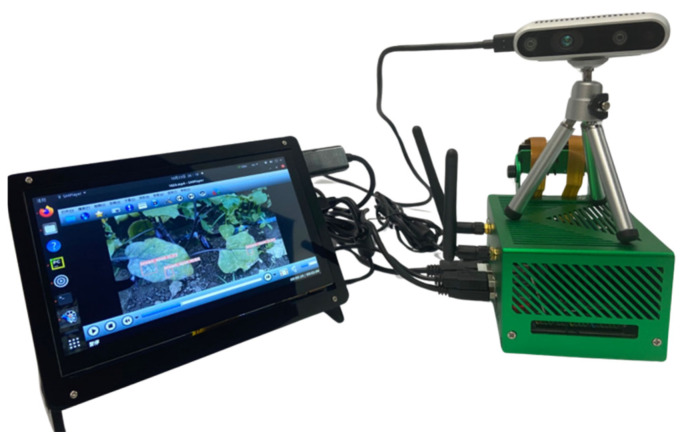
Edge detection equipment using different detection models to identify disease in eggplant leaves.

**Figure 21 plants-13-02303-f021:**
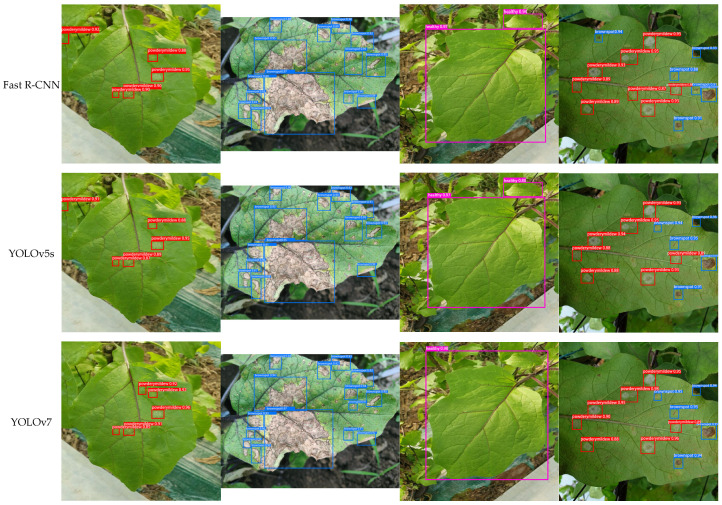
The detection results of eggplant leaves by different models were the (**a**) detection results of powdery mildew with different models; (**b**) detection results of brown spot disease by different models; (**c**) detection results of healthy leaves by different models; and (**d**) results of different models.

**Figure 22 plants-13-02303-f022:**
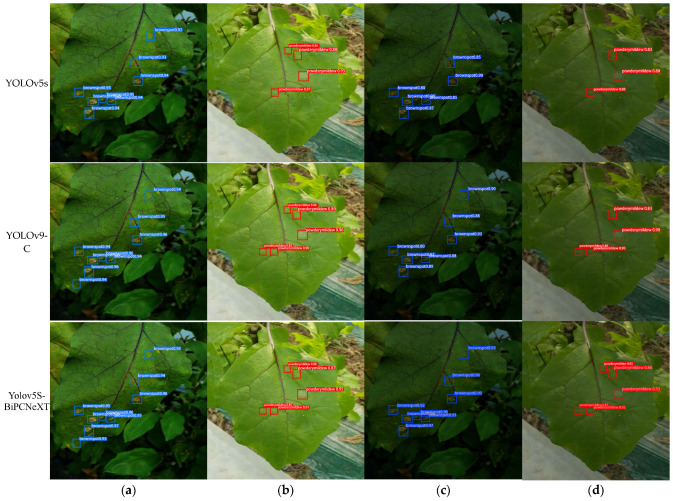
According to the detection results of different models for diseased eggplant leaves under different illumination, (**a**,**b**) are brown spot and powdery mildew photos under original illumination and (**c**,**d**) are brown spot and powdery mildew photos under dark light.

**Table 1 plants-13-02303-t001:** Different leaf distribution conditions in training.

Leaf Category	Number of Labels before Augmentation	Increased Number of Labels
Brown spot disease	303	1890
Powdery mildew	188	1175
Healthy	293	1468
Totality	784	4533

**Table 2 plants-13-02303-t002:** Experimental hardware environment configuration.

Configuration Name	Environmental Parameter
Operating system	Windows10(64-bit professional version)
CPU	13th Gen Intel(R) Core(TM) i5-13600KF
GPU	NVIDIA GeForce RTX 4070 Ti, Graphics memory is 12GB
RAM	32GB
Deep learning framework	Pytorch-GPU2.1.0+cu118
GPU acceleration library	cuDNN8.9.5

**Table 3 plants-13-02303-t003:** Comparison of the test results between YOLOv5s and YOLOv5s-BiPCNeXt.

Disease	Models	P (%)	R (%)	mAP@_0.5_ (%)	mAP@_0.5:0.95_ (%)	Parameter	GFLOPs	Weights(MB)
All	YOLOV5s	95.9	93.1	94.3	71.9	7,043,864	15.8	13.6
Healthy		98.5	98.3	98.8	91.0			
Brown spot		94.5	90.7	92.5	62.6			
Powdery mildew		94.7	88.8	91.6	62.1			
All	YOLOv5s-BiPCNeXt	96.2	94.8	96.5	75.8	4,151,808	9.8	8.8
Healthy		99.4	99.4	99.5	94.4			
Brown spot		94.8	92.7	94.9	66.1			
Powdery mildew		94.5	92.4	95.0	67.0			

**Table 4 plants-13-02303-t004:** Results of ablation experiments.

MobileNeXt	C3-BiPC	CARFE	EMA	mAP@_0.5_ (%)	Parameters	GFLOPs
-	-	-	-	94.3	7,043,864	15.8
√				91.5	4,836,435	10.5
-	√	-	-	96.0	6,565,575	14.3
-	-	√	-	94.5	6,974,363	16.0
√	√	-	-	94.8	5,869,478	10.8
√	√	√	-	95.5	5,435,347	10
√	√	√	√	96.5	4,151,808	9.8

**Table 5 plants-13-02303-t005:** Contrast of different attention mechanisms.

Attention Mechanism	P (%)	R (%)	${mAP@}_{0.5}$(%)	Parameters	GFLOPs
CA	93.2	92.3	94.5	5,043,864	10.8
SE	95.5	92.5	93.2	4,973,252	11.0
CBAM	94.4	93.3	95.0	5,368,654	14.3
EMA	96.2	94.8	96.5	4,151,808	9.8

**Table 6 plants-13-02303-t006:** Performance comparison of mainstream detection models.

Models	P (%)	R (%)	mAP@0.5 (%)	Parameters	GFLOPs	Weights (MB)	FPS
Fast R-CNN	74.3	85.6	95.8	22,485,357	303.6	108.3	8
YOLOv5s	95.9	93.1	94.3	7,043,864	15.8	13.6	15
YOLOv7	93.6	93.2	96.4	33,556,879	315.5	120.8	10
YOLOv8s	89.8	95.2	94.5	10,394,568	27.4	30.1	15
YOLOv9-C	96.2	95.5	97.2	51,004,178	238.9	102.8	13
YOLOv5-BiPCNeXt	96.2	94.8	96.5	4,151,808	9.8	8.8	26

## Data Availability

The data presented in this study are available on request from the corresponding author.

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
