# Peer review of "YOLOv5s-BiPCNeXt, a Lightweight Model for Detecting Disease in Eggplant Leaves"

_plants, 2024, doi:10.3390/plants13162303_

Round 1
Reviewer 1 Report
Comments and Suggestions for Authors
The manuscript addresses the issue of eggplant leaf disease detection, and the challenge is to provide good accuracy while maintaining a lightweight computing footprint. To achieve this, the classification models need to be highly accurate while consuming minimal computational resources, allowing for implementation on mobile or edge devices that can be deployed in the field.
The authors' solution is based on a number of elements, such as the MobileNeXt backbone, C3-BiPC module, CARAFE upsampling, and the EMA attention mechanism.
An ablation study validates the effectiveness of these components, and the experimental results show that the model can achieve high classification performance while reducing the number of parameters and associated computational costs compared to YOLOv5s as well as other models.
These contributions are significant, and overall, the article is well-structured and written, however, there are a few aspects that should be revised: The authors should clarify the original contributions. For example, they should properly cite the components used, such as the CARAFE operator, EMA attention mechanism, MobileNeXt backbone, as well as the elements that are used in the C3-BiPC module, including the Biformer attention mechanism, etc.
An error analysis would be helpful to understand which types of images the classifier performs poorly on, allowing the identification of ways to improve the solution.
Comments on the Quality of English Language
The English language is mostly fine, but some editing is still required.
Author Response
请参阅附件。

Reviewer 2 Report
Comments and Suggestions for Authors
The paper describes a method to detect eggplant leaf diseases to ensure healthy growth and increase productivity. It proposes a lightweight detection model, YOLOv5s-BiPCNeXt, which includes a lightweight backbone network to reduce the number of parameters and a multi-scale cross-space attention mechanism and a content-aware feature recombination operator instead of the nearest neighbor interpolation algorithm to effectively improve the spatial resolution of the disease feature map. Compared to existing models, this approach offers higher detection accuracy and lower computational costs. The experimental results show very good average accuracies for detecting several diseases and excellent for healthy leaves. This approach is compatible with a real-time implementation on light devices.
The context, approach design and research results are clearly presented. The conclusions are in coherence with the results.
Presentation and style issues:
- Lot of space missing after comas, parenthesis/ and dots (mainly dots)
- Adopt the same choice for space/not in all references: xxx[i] or xxx [i]
- Pay attention to width of illustrations with respect to text one.
- Uppercases at the start of the line missing (table 1)
Reviewer 3 Report
Comments and Suggestions for Authors
The paper presents the development of YOLOv5 based lightweight detection algorithms to reduce possible measurement error or computation cost in a better natural condition way. The point I perceive as most important from this paper is to reduce the computational complexity and number of parameters for small disease targets detection and its connections required for real-time operation. Mathematical deduction process is not rigorous. Most mathematical content description in this paper is rough and not consistence. Typo errors (which impede readability) or format occur in many lines of the manuscript. Many content descriptions are redundant or unnecessary. The technical depth of the paper was appropriated for the generally knowledgeable individual working in the deep learning or a related field. As a research manuscript, this paper shows a lack of demonstration of theory and reasonable field verification and comparison. This includes the optimization process of C3-BiPC and the unrigorous experimental process.
Abstract
The abstract had to be rewritten, and the original content was full of technical terms that only people familiar with the field of AI could understand. This content does not allow people in the field of “plant medicine” to get help and understanding. The content lacks description of key technologies or optimization technologies for small disease targets or lightweighting. It is recommended to write briefly according to the following principles: introduction - problem - method (including innovation + complexity) - results (limitations and performance improvement effects).
Introduction
The author introduces many methods for detecting diseases, including machine learning, color space conversion and deep learning techniques (HSI, SVM, CNN, YOLO, etc.). However, this does not substantially reflect the previous literature on the key technologies of this study, including lightweight, requires less computation, and detects tiny objects. It is recommended that the author delete unnecessary documents and list the documents that are highly relevant to this study for citation.
Secondly, the author mentioned natural conditions to detect diseases in the abstract. Why not list more literature in this area to emphasize that this method is unique.
Line 149-152: This sentence has no basis and cannot reflect the actual problem. This requires citing some important documents.
Line 1157-159: This passage does not reflect the content described in line 149-152. This needs to be corrected and some references need to be cited.
Line 163-165: The test object needs to be specifically defined, including the scientific name of the disease.
Method and Material
Section 2.1-2.2: Content describing images of infected disease categories and the source of the images. At the same time, the data set also includes image amplification images.
Section 2.3: This content is more suitable for readers in the field of deep learning. The reasoning process is not smooth and he lacks some explanations. The content does not provide those interested readers or researchers with the ability to reproduce the results.
Figure 6: This diagram is unclear, and it is not stated what operations the blocks within the diagram represent. This does not allow the reader to understand the approach.
Figure 7: If this study focuses on lightweight model design, this figure should be described in detail and compared with Figure 6. It is recommended to highlight the differences between these lightweight structures and the original ones, and list the mathematical inference process is needed. In addition, the description of EMA, CARAFE, and C3-BiPC is unclear. It seems to be just a conceptual introduction. Please clarify or cite relevant literature.
Page 9, line 290-294: This paragraph cannot represent anything, and the content does not have any scientific basis or result verification.
Page 9, line 305-306, please confirm the correlation between equations (1) and (2).
Line 360-363: This paragraph requires documentary support.
Line 366-268: This description is redundant or meaningless.
Line 402: FLOP full name is required.
Page 11-15: The content presented is difficult to follow, which makes it difficult for readers to understand the method of causing pests and diseases. It is recommended to optimize mathematical symbols and fully define some symbols, such as "*", "", "-". In addition, many operation function bit definitions, such as gather(), topkIndex(), Attention()? Please explain (13)-(17) in detail, especially its connection with Figure 15 and the optimization process. Figure 16: The abbreviations in the boxes in the figure need to be defined.
Results and Discussion
Although the experimental results are in line with expectations, however, the content presented cannot reflect the methodology, especially the definition of parameters, setting conditions, and estimation methods of weights. Why use GFLOPS to evaluate performance instead of using FLOPS provided by (11), (12)?
In comparing different methods, such as Fast R-CNN or YOLOv9-C, how to estimate the amount of Parameters, GFLOPs and Weights of the above methods?
Line 549, 563: Repeated definition of the full name of GFLOP.
Line 575: How to obtain or estimate GFLOP ”8.8 MB”
Table 3: Are these performance comparison results conducted under the same conditions for each method to be implemented? Please clarify and describe how Parameters, GFLOPs, and Weights are estimated.
Figure 17: You present a good comparison of results. However, this result cannot show the comparison of the small object detection performance of this method. What does Figure 20 not represent? Describe the recognition effect of this method under different light source conditions, including analysis and comparison of statistical results.
Conclusions
The conclusion cannot reflect the results presented in the content. For example, the description of the process of implementing the optimization module (C3-BiPC) and CARAFE upsampling operator is unclear. What is "optimization"? Please clarify.
Others
The format of literature review is not consistent, especially the title of paper. These are a selection, there are loads more, please check again.
Comments on the Quality of English LanguageNo comments.
Reviewer 4 Report
Comments and Suggestions for Authors
This paper proposes the YOLOv5s-BiPCNeXt detection model for eggplant / aubergine leaf diseases. The document is generally well organised and easy to follow. In my opinion, the first section could have been divided into 1. Introduction and 2. State of the art. The description of the model is detailed in depth and the validation includes the comparison with the original version of the model, an ablation study and newer versions of the model.
Please see my comments below:
- Page 4: (ll. 157-158) "Therefore, given the limited computational abilities of edge detection devices". Please, describe what you consider to be an "edge detection device". You can refer to Figure 19.
- Page 5, Figure 3: Include a description of each image feature in the caption.
- Page 6 (ll. 228-230): "The 980 original photos collected were divided (...) according to the ratio of 8:1:1 using Python processing tool". Please justify this division and include the corresponding reference since, as far as I know, this division is not common.
- Increase the size of Figures 6 and 7 for readability.
- Check equations (5)-(12). In some cases, c and cp are in upper case, in others in lower case, but it seems to be the same value.
- Page 21, line 659. Heading text is shown in bold. It should be in italics, not bold.
Finally, check for some minor errors:
- Some words appear together:
+ Page 1: "network.Additionally,the"; "mul-tiscale"; "map.Experimental"; "equirement.In"; "dis-eases"; "As shown in the 41 Figure 1, A particularly".
+ Page 2: "yield.Another"; "yield.Thus"; "90%.However"
+ Page 3: "accuracy.Zhou[21]"; "applications.In"; "DF-SSD.It"; "91.07%.Liu"; "scenarios.Li".
+ This is repeated on the following pages...
- Page 4: (ll. 151-152) "achieev".
- Page 6, Figure 4: "(b) random occlusion" --> "(b) random occlusion + flip horizontal";
"(c) flip horizintal" --> "(c) flip horizontal"
"(e) rotation+adjusted brightness" --> "(e) adjusted brightness"
- Page 6 (ll. 222-223). Check the following sentence: "Add the tag box and generate the corresponding label file for the area of eggplant leaf disease in the image.".
- Page 7, "Table 1. Different leaf distribution conditions" --> "Table 1. Different leaf distribution conditions in training".
- Page 11, line 406. Check this sentence: "When the input and output images size, the formula changes to".
- Page 14, line 466: ", The" --> ", the".
- Page 19, line 614: "mecha-nisms" --> "mechanisms".
Reviewer 5 Report
Comments and Suggestions for Authors
This study proposes a YOLOv5s-BiPCNeXt lightweight detection model to detect eggplant leaf disease effectively.
The introduction frames the problem by considering real-life situations and performs an excellent state-of-the-art analysis.
The methodology is well explained and characterized. The results validate it as promising and solid. A good comparison is performed with the state-of-the-art.
Major Concerns:
(1) The manuscript needs an in-depth review of grammar, punctuation, and sentence structure.
(2) The Introduction, a crucial part of the manuscript, should end with a statement on the significant content that is going to be presented in the paper, a set of claims, and a concise summary of the remaining paper.
(3) Section 2 needs some review. The method should come first, as is the natural order of deep learning proceedings: Method, Dataset, and Results.
(4) Section 3.1 (first paragraph) must justify why the settings are chosen this way. Otherwise, a sensibility analysis must be performed.
(5) Claims appear for the first time in the conclusion. A brief should be included in the introduction, and a very brief should be included in the abstract.
Minor concerns:
(1) use a space between the last word and a reference number. Most of them do not have. Please correct it.
(2) There are many typos regarding punctuation. Example: Line 81 "exceeding 90%.However", Line 102 "racy.Zhou[21]".
(3) Please correct sentence structure. Example: Line 100 "He[20] proposed the MFaster R-CNN algorithm". Line 332 "diseases.To address". Line 403 "Where,𝑘𝑤and𝑘ℎrepresent", Line 459 " different mod-els in real", Line710 "4.conclusions", among many others!
Comments on the Quality of English LanguageThe manuscript needs an in-depth review of grammar, punctuation, and sentence structure.
Round 2
Reviewer 3 Report
Comments and Suggestions for Authors
The authors need to describe in more depth the novelty of this work and how the impact of deep learning techniques on the specific biological applications studied was performed, manipulated, and/or analyzed.
